# DEEP INTERACTION PROCESSES FOR TIME-EVOLVING GRAPHS

## ABSTRACT

Time-evolving graphs are ubiquitous such as online transactions on an e-commerce platform and user interactions on social networks. While neural approaches have been proposed for graph modeling, most of them focus on static graphs. In this paper we present a principled deep neural approach that models continuous time-evolving graphs at multiple time resolutions based on a temporal point process framework. To model the dependency between latent dynamic representations of each node, we define a mixture of temporal cascades in which a node's neural representation depends on not only this node's previous representations but also the previous representations of related nodes that have interacted with this node. We generalize LSTM on this temporal cascade mixture and introduce novel time gates to model time intervals between interactions. Furthermore, we introduce a selection mechanism that gives important nodes large influence in both $k-$depth subgraphs of nodes in an interaction. To capture temporal dependency at multiple time-resolutions, we stack our neural representations in several layers and fuse them based on attention. Meanwhile, our method can process unseen nodes and their interactions. Experimental results on interaction prediction and classification tasks – including a real-world financial application – illustrate the effectiveness of the time gate, the selection and fusion mechanisms of our approach, as well as its superior performance over the alternative approaches.

## 1 INTRODUCTION

Representation learning over graph data has become a core machine learning task with a wide range of applications including e-commerce, finance, social networks, and bioinformatics. Various neural graph representations such as (Perozzi et al., 2014; Grover & Leskovec, 2016; Wang et al., 2016; Kipf & Welling, 2017; Defferrard et al., 2016; Scarselli et al., 2009; Ying et al., 2018; Hamilton et al., 2017b; Monti et al., 2017; Den Berg et al., 2017) have been proposed to learn from *static* graph data and successfully used for downstream tasks (*e.g.*, classification). Graph data, however, are often dynamic in practice; nodes and interactions between them can grow and shrink. A straightforward approach to handle dynamic graphs is to compress them into one or several static graphs. The drawbacks of this approach are multifold; we not only blur temporal structural information but also miss time information that can be critical for real-world applications. An illustrative example is given in figure 1.

To handle continuous time-evolving graph, we can approximate a by a sequence of snapshot graphs, each of which includes all interactions that occur during a user-specified discrete-time interval, as shown in (Goyal et al., 2018; Leskovec et al., 2007; Zhou et al., 2018; Sankar et al., 2019). This treatment reduces time resolution and it is tricky to specify the appropriate aggregation granularity. To avoid these problems, Nguyen et al. (2018) proposed continuous-time dynamic networks (CTDNE) that generalize deep walk methods to learn time-dependent network embedding. As a *transductive* method, CTDNE cannot handle the growth of new nodes. Dai et al. (2016) applied temporal point processes to model time-evolving graphs and, as a nonparametric Bayesian approach, their approach can naturally cope with the growth of new nodes and interactions. They used recurrent neural networks (RNNs) to define an intensity function in temporal point processes. These RNN models are shallow and one-step unrolled, making it easy to compute but relatively limited in modeling power. Trivedi et al. (2019) extended this approach by modeling two-time scale and adopting temporal-attention mechanism.

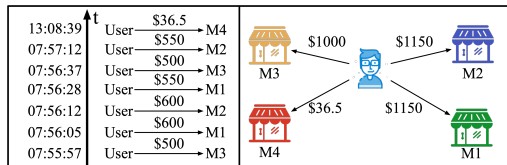 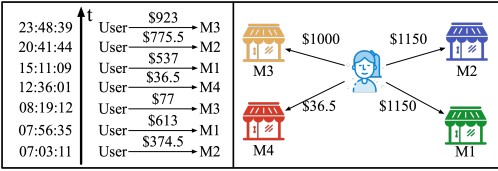

(a) An illegal cash-out event                    (b) Legal transactions

Figure 1: An illustrative example. Figure (a) shows an illegal cash-out event. It can be revealed by high-frequency transactions with multiple merchants. However, if we merge the transaction data into a static graph, we cannot distinguish it from the static graph generated from normal online shopping activities. Thus, learning from such a static graph will fail to detect the cash-out event.

In this paper we present a powerful deep neural approach that models continuous time-evolving graphs at multiple time resolutions based on a temporal point process framework. We name the new approach *deep interaction processes* (DIPs). To model the dependency between latent dynamic representations of each node, we define a mixture of temporal cascades in which a node's neural representation depends on not only this node's previous representations but also the previous representations of related nodes that have interacted with this node. We generalize LSTM on this temporal cascade mixture and introduce novel time gates to model time intervals between interactions. Furthermore, We introduce a selection mechanism that gives important nodes large influence in both $k-$depth subgraphs of nodes in an interaction. To obtain representations from fine-to-coarse time-resolutions, we stack our neural representations in several layers and fuse them based on attention. Based on the temporal point process framework, our approach can naturally handle growth of graph nodes and interactions, making it inductive.

The rest of the paper is organized as follows. In Section 2 we give background on temporal point processes and in Section 3 we present the new DIP approach. In Section 4 we discuss related works. In Section 5 we report experimental results on multiple interaction prediction and classification tasks including an important real-world anti-fraud financial application, demonstrating superior performance of the new approach over the alternatives.

## 2 TEMPORAL POINT PROCESSES

We first describe temporal point processes (a class of nonparametric Bayesian models) that our approach is based on. Specifically, a temporal point process is a stochastic process that generates a sequence of discrete events localized at times $\{t_i\}_{i=1}^{N}$ in any given observed time window $[0, T]$, where $N$ is the number of events. An important way to characterize temporal point processes is via the conditional intensity function $\lambda(t|H_t)$ -the stochastic model for the next event time $t$ given all historical events before time $t$, denoted as $H_t = \{t_i | t_i < t\}$. Formally, within a small time window $[t, t + dt)$, $\lambda(t|H_t) dt$ is the probability for the occurrence for a new event given the $H_t$: $\lambda(t|H_t) dt = P\{ \text{event in } [t, t+dt) | H_t\}$. From the survival analysis theory(Aalen et al., 2008), given the times of the past events $\{t_1, t_2, \ldots, t_i\}$, the conditional density that an event occurs at $t_{i+1}$ is given as follows:$p\left(t_{i+1}|H_{t_{i+1}}\right) = \lambda\left(t_{i+1}|H_{t_{i+1}}\right) \exp\left\{ - \int_{t_i}^{t_{i+1}} \lambda(t|H_t) dt \right\}$,where the exponential part in the above equation means the conditional probability that no event happens during $[t_i, t_{i+1})$. The functional forms of the conditional intensity function $\lambda(t|H_t)$ can represent certain forms of dependencies of the historical events. For instance, for Poisson processes(Kingman, 2005) we set $\lambda$ to be constant – making the assumption that the process is stationary and the temporal events in history are independent of each other. For classical Hawkes processes(Hawkes, 1971), the intensity function $\lambda$ is often set to be a sum of multiple exponential functions, assuming that the mutual excitation among events is positive, additive over the past events, and exponentially decaying with time. Mei & Eisner (2017a) removed these limiting assumptions using LSTM to learn $\lambda$ from data.

## 3 DEEP INTERACTION PROCESSES

In this section, we present a new neural non-parametric Bayesian approach over continuous-time evolving graphs. First, we present a temporal dependency graph that is a mixture of the temporal cascades, to model interdependence between graph nodes (as well as latent node representations).

Then we present a novel deep model to learn dynamic node representations in the temporal dependency graph. This model naturally generalizes a chain-structured LSTM to a temporal Graph-structured LSTM equipped with time gates to handle interaction with irregular time intervals. Furthermore, we propose the FUSION and SELECTION methods to enhance the dynamic interactive nodes representation. Given the dynamic node representations, we define deep interaction processes that model potential interactions between any two nodes over time and we finally layout the maximum likelihood estimation method to optimize it. A toy example is shown in Figure 2 to introduce the corresponding concepts and the whole procedure of computing enhanced dynamic representation is given in Figure 3.

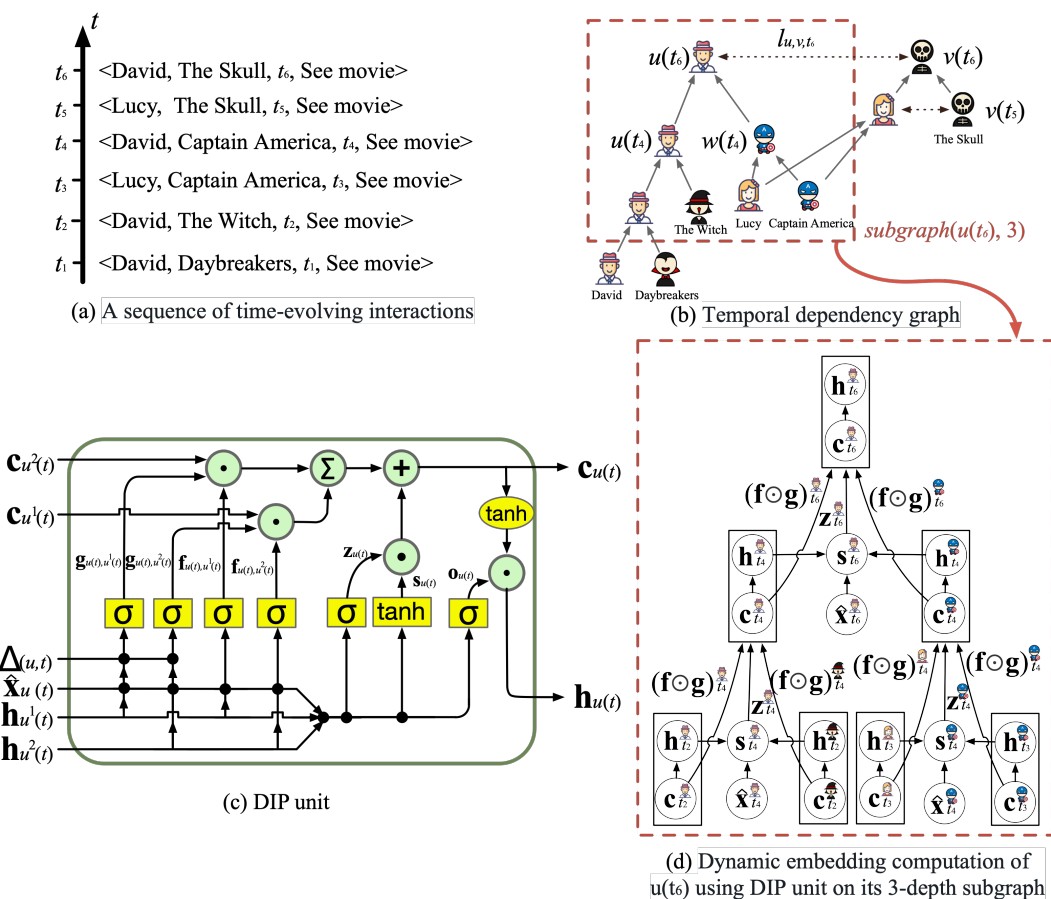

Figure 2: A toy example of interactions, the corresponding temporal dependency graph and computation of nodes' dynamic embedding using DIP unit on its 3-depth temporal dependency subgraph.

### 3.1 TEMPORAL DEPENDENCY GRAPH

Consider a collection of people-movie interactions at different time points (*e.g.*, David saw the movie The Skull at $t_6$.) as shown in figure 2(a). The people and movies form a temporal dependency graph in which each person or movie is a node and interactions happen over time (in figure 2(b)). After one interaction occurs, we update the neural representation of the two nodes linked to this interaction; *e.g.*, right after time $t_6$, we update the representations for David and movie The Skull. The new neural representation of David depends on both his current and previous interactions – as a result, depending on the representations of the two nodes associated with the previous interaction. This naturally forms a dependency cascade. Similarly we can obtain a dependency cascade for Lucy's representations. Because of the common movies David and Lucy saw, their dependency cascades overlap and form a cascade mixture. Formally, we denote a dynamic interaction or link at time $t$ by $l_{u,v,t}$ where $u$ and $v$ are two nodes associated with this interaction. We denote the node $u$ at time $t$ by $u(t)$ and the two nodes associated with $u$'s precedent interaction at time $t^-$ as $u^1(t)$ and $u^2(t)$. For later usage,

we denote the $k-$depth temporal dependency subgraph for $u(t)$ as *subgraph*$(u(t), k)$ as shown in figure 2(b).

## 3.2 DIP NEURAL UNIT AND DYNAMIC REPRESENTATION

Now we present a novel neural unit to update dynamic latent representations of nodes over the temporal dependency graph which is illustrated in figure 2(c) and figure 2(d).

First, let us denote node $u$'s features or embedding (i.e., a static representation jointly learned from data) at time $t$ by $\mathbf{x}_{u(t)}$ and denote features of interaction $l$ by $\mathbf{x}_l$. The interaction feature can be empty if the interaction contains only the temporal information. The concatenation of $\mathbf{x}_{u(t)}$ and $\mathbf{x}_l$ is denoted by $\hat{\mathbf{x}}_{u(t)}$. Let $\Delta_{(u,t)} = t - t^-$ be the time interval between two consecutive interactions involving $u$ at time $t$ and $t^-$. Our neural unit, i.e, DIP unit, generalizes a chain-structured LSTM unit to depict the temporal dependency on graph data. The DIP unit has an update gate $\mathbf{s}$, an input gate $\mathbf{z}$, an output gate $\mathbf{o}$ and two forget gates $\mathbf{f}$ over $\hat{\mathbf{x}}_{u(t)}$, dynamic representation of $u^i(t)$, i.e, $\mathbf{h}_{u^i(t)}$ and cell states $\mathbf{c}_{u^i(t)}$ $(i = 1, 2)$ as shown in figure 2(c). Additionally, we introduce time gates $\mathbf{g}$ to capture the impacts of irregular time interval $\Delta_{(u,t)}$. Specifically, $\mathbf{h}_{u(t)}$ and $\mathbf{c}_{u(t)}$ are updated as follows:

$$\mathbf{z}_{u(t)} = \sigma \left( \mathbf{W}_z \hat{\mathbf{x}}_{u(t)} + \sum_{i=1}^{N} \mathbf{R}_{z_i} \mathbf{h}_{u^i(t)} + \mathbf{b}_z \right) \qquad \mathbf{o}_{u(t)} = \sigma \left( \mathbf{W}_o \hat{\mathbf{x}}_{u(t)} + \sum_{i=1}^{N} \mathbf{R}_{o_i} \mathbf{h}_{u^i(t)} + \mathbf{b}_o \right)$$

$$\mathbf{s}_{u(t)} = \tanh \left( \mathbf{W}_s \hat{\mathbf{x}}_{u(t)} + \sum_{i=1}^{N} \mathbf{R}_{s_i} \mathbf{h}_{u^i(t)} + \mathbf{b}_s \right) \quad \mathbf{f}_{u(t),u^i(t)} = \sigma \left( \mathbf{W}_{f_i} \hat{\mathbf{x}}_{u(t)} + \mathbf{R}_{f_i} \mathbf{h}_{u^i(t)} + \mathbf{b}_{f_i} \right)$$

$$\mathbf{g}_{u(t),u^i(t)} = \sigma \left( \mathbf{W}_{g_i} \hat{\mathbf{x}}_{u(t)} + \mathbf{R}_{g_i} \mathbf{h}_{u^i(t)} + \mathbf{M}_{g_i} \Delta_{(u,t)} + \mathbf{b}_{g_i} \right)$$

$$\mathbf{c}_{u(t)} = \mathbf{z}_{u(t)} \odot \mathbf{s}_{u(t)} + \sum_{i=1}^{N} \mathbf{f}_{u(t),u^i(t)} \odot \mathbf{c}_{u^i(t)} \odot \mathbf{g}_{u(t),u^i(t)}$$

$$\mathbf{h}_{u(t)} = \mathbf{o}_{u(t)} \odot \tanh \left( \mathbf{c}_{u(t)} \right) \tag{1}$$

where $\sigma, \tanh$ and $\odot$ represent the sigmoid function, the hyperbolic tangent function, and the Hadamard product (pointwise multiplication), respectively, and parameters in the unit including the recurrent weights $\mathbf{R}_{z_i}$, $\mathbf{R}_{o_i}$, $\mathbf{R}_{s_i}$, $\mathbf{R}_{f_i}$ and $\mathbf{R}_{g_i}$, the projection matrices $\mathbf{W}_z$, $\mathbf{W}_o$, $\mathbf{W}_s$, $\mathbf{W}_{f_i}$ and $\mathbf{W}_{g_i}$, the bias vectors $\mathbf{b}_z$, $\mathbf{b}_o$, $\mathbf{b}_s$, $\mathbf{b}_{f_i}$ and $\mathbf{b}_{g_i}$ and the time weight matrix $\mathbf{M}_{g_i}$ are learned from data. For convenience, we use DIP-UNIT $(\cdot)$ to summarize the above equations, then the dynamic representation of node $u(t)$ is given as follows:

$$\mathbf{h}_{u(t)}, \mathbf{c}_{u(t)} = \text{DIP-UNIT} \left( \hat{\mathbf{x}}_{u(t)}, \mathbf{c}_{u^1(t)}, \mathbf{c}_{u^2(t)}, \mathbf{h}_{u^1(t)}, \mathbf{h}_{u^2(t)}, \Delta_{(u,t)}, \Theta \right) \tag{2}$$

where $\Theta$ represent all the parameters.

Considering the computational cost in practice, when obtaining nodes' dynamic representation given an happened interaction, we only utilize history information in the $subgraph(u(t), k)$ as illustrated in figure 2(d), which is similar to a chain-structured LSTM training unfolded with the max k steps. The k is a hyper-parameter.

## 3.3 ENHANCED DYNAMIC REPRESENTATION

### 3.3.1 STACKING AND FUSION

To model nonlinear dependency relationships at different temporal resolutions, we stack $L$ layers of DIP-UNIT together. The output of the $j$-th layer is computed recursively as follows:

$$(\mathbf{h}_{u(t)}^j, \mathbf{c}_{u(t)}^j) = \text{DIP-UNIT}^j \left( \mathbf{h}_{u(t)}^{j-1}, \mathbf{c}_{u^1(t)}^j, \mathbf{c}_{u^2(t)}^j, \mathbf{h}_{u^1(t)}^j, \mathbf{h}_{u^2(t)}^j, \Delta_{(u,t)}, \Theta_j \right) \tag{3}$$

were $\mathbf{h}_{u(t)}^0 = \hat{\mathbf{x}}_{u(t)}$, $j = 1, \ldots, L$. To train deeper dynamic neural networks easily, we employ the residual connection as the following form: $skip(\mathbf{h}_{u(t)}^{j-1}, \mathbf{h}_{u(t)}^j) = \mathbf{W}_{\mathbf{skip}} \mathbf{h}_{u(t)}^{j-1} + \mathbf{h}_{u(t)}^j$ where $\mathbf{W}_{\mathbf{skip}}$ is a weight matrix. Motivated by *ELMo* (Peters et al., 2018), we fuse all internal dynamic representations from all the layers to achieve rich dynamic representations. The fusion is a weighted summation of all layers defined as follows: $\mathbf{h}_{u(t)} = \gamma \sum_{j=0}^{L} \alpha_j \mathbf{h}_{u(t)}^j$, where $\alpha_j$ are softmax-normalized weights and $\gamma$ is a scaling parameter. They both are learned parameters.

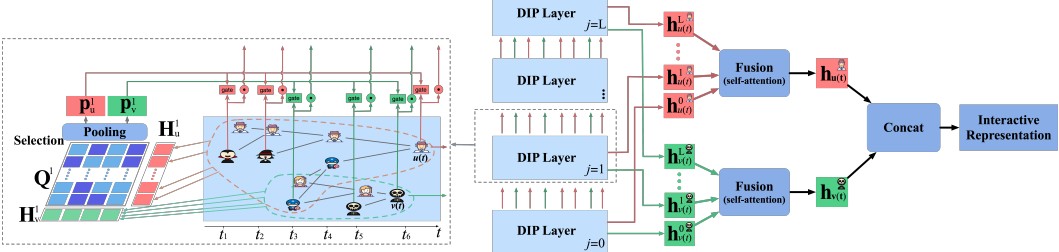

Figure 3: An illustrative example. Enhanced dynamic representation with Fusion and Selection.

### 3.3.2 SELECTION

Given an interaction $l_{u,v,t}$, it is reasonable to assume that not all the historical interactive nodes have the equal importance for formalizing this interaction. Thus we use a two-phase gating mechanism to *select* relevant nodes to learn dynamic representations and cell states of the current node. Specifically, a co-attention mechanism is first used to measure relevance of historical time-evolving patterns between $subgraph(u(t), k)$ and $subgraph(v(t), k)$,

$$\mathbf{Q}^j = \tanh\left(\mathbf{H}_u^{j\top}\mathbf{W}_Q\mathbf{H}_v^j\right) \tag{4}$$

where $\mathbf{H}_u^j = \left[\mathbf{h}_1^j, \ldots, \mathbf{h}_a^j, \ldots, \mathbf{h}_m^j\right], a \in subgraph(u(t), k)$, $\mathbf{H}_v^j = \left[\mathbf{h}_1^j, \ldots, \mathbf{h}_e^j, \ldots, \mathbf{h}_n^j\right], e \in subgraph(v(t), k)$, $m$ and $n$ are the numbers of nodes in the two corresponding $subgraphs$, $\mathbf{W}_Q \in \mathbb{R}^{d \times d}$ are the weight parameters. The $\mathbf{Q}^j$ is a co-attention affinity matrix which captures the relevance information in $subgraph(u(t),k)$ and $subgraph(v(t),k)$. The co-dependent global embedding $\mathbf{p}_u^j$, $\mathbf{p}_v^j$ are obtained by the following equations.

$$\mathbf{p}_u^j = \mathbf{H}_u^j \text{SoftMax}\left(\underset{ColWise}{\text{Max}}\ \mathbf{Q}^j\right) \qquad \mathbf{p}_v^j = \mathbf{H}_v^j \text{SoftMax}\left(\underset{RowWise}{\text{Max}}\ (\mathbf{Q}^j)\top\right) \tag{5}$$

where Max means max-pooling operation which is used to choose the most relevant information for the maximum influence (or affinities) on nodes in the corresponding subgraph. In addition, to adjust the importance of historical nodes, two adaptive gate functions are designed for nodes in $subgraph(u(t),k)$ and $subgraph(v(t),k)$ respectively,

$$g_u(\mathbf{p}_u^j, \mathbf{h}_a^j) = \sigma(\mathbf{w}_p\mathbf{p}_u^j + \mathbf{w}_h\mathbf{h}_a^j) \qquad g_v(\mathbf{p}_v^j, \mathbf{h}_e^j) = \sigma(\mathbf{w}_p\mathbf{p}_v^j + \mathbf{w}_h\mathbf{h}_e^j) \tag{6}$$

where the weights $\mathbf{w}_p$, and $\mathbf{w}_h$ are shared by all the stacked layers. Using these gates, we enhance the dynamic node representations as follows:

$$(\mathbf{h}_{u(t)}^j, \mathbf{c}_{u(t)}^j) = \text{DIP-UNIT}^j\left(\mathbf{h}_{u(t)}^{j-1} \odot g_u(\mathbf{p}_{u(t)}^{j-1}, \mathbf{h}_{u(t)}^{j-1}), \mathbf{c}_{u^1(t)}^j, \mathbf{c}_{u^2(t)}^j, \mathbf{h}_{u^1(t)}^j, \mathbf{h}_{u^2(t)}^j, \Delta_{(u,t)}, \Theta_j\right) \tag{7}$$

Similarly, we can compute $(\mathbf{h}_{v(t)}^j, \mathbf{c}_{v(t)}^j)$ for $v(t)$ based on the selection mechanism.

### 3.4 CONDITIONAL INTENSITY FUNCTION

We model the dynamic interactions as a multi-dimensional temporal point process. Specifically, we define the conditional intensity function of the temporal point process at the dimension indexed by $(u, v)$, given its graph-structured history $H_t^{u,v}$ where $H_t^{u,v} = subgraph(u^1(t), k) \cup subgraph(u^2(t), k) \cup subgraph(v^1(t), k) \cup subgraph(v^2(t), k)$, as follows:

$$\lambda^{u,v}(t|H_t^{u,v}) = SoftPlus\left(\boldsymbol{h}_t^{u,v}\mathbf{w}_\lambda + \mathbf{w}_t^\top\tau + b_\lambda\right) \tag{8}$$

where $\boldsymbol{h}_t^{u,v} = \left[\mathbf{h}_{u^1(t)}^\top, \mathbf{h}_{u^2(t)}^\top, \mathbf{h}_{v^1(t)}^\top, \mathbf{h}_{v^2(t)}^\top\right], \tau = \left[\Delta_{(u,t)}, \Delta_{(v,t)}\right]^\top$, the scalar $b_\lambda$ can be viewed as a base intensity level for the occurrence of the next interaction, and the *SoftPlus* function is used to ensure the non-negativity of the intensity. A key step for obtaining $H_t^{u,v}$ is to get the $k$-depth *subgraphs* of $u$ and $v$'s direct dependants. Please see **Appendix.A** for more details about fast obtaining $k$-depth *subgraphs*.

### 3.5 MAXIMUM LIKELIHOOD PARAMETER ESTIMATION

#### 3.5.1 INTERACTION PREDICTION

Given a set of interactions as $I = \{(u_i, v_i, t_i)\}_{i=1}^{i=N}$ observed in a time window [0, T], we can learn the model by minimizing the negative joint log-likelihood of $I$ as follows: $\mathcal{L}_1 = -\sum_i \log P^{u_i,v_i}\left(t_i | H_{t_i}^{u_i,v_i}\right)$ where $P^{u_i,v_i}(t_i | H_{t_i}^{u_i,v_i})$ represents the probability of formalizing an interaction between $u_i$ and $v_i$ at time $t_i$ given the dependant history of non-chain structures $H_{t_i}^{u_i,v_i}$. Based on the intensity definition, we have $\mathcal{L}_1 = -\sum_i \log \lambda^{u_i,v_i}\left(t_i | H_{t_i}^{u_i,v_i}\right) + \int_0^T \Lambda(t)dt$, where $\Lambda(t) = \sum_{u,v} \lambda^{u,v}(t)$. Since the survival part does not have an analytic solution, we apply Monte Carlo to do numerical integrations. We follow the negative sampling approaches used by Dai et al. (2016) and Trivedi et al. (2019) to accelerate the survival term calculation.

#### 3.5.2 INTERACTION CLASSIFICATION

An interaction sequence with markers is denoted as $I' = \{(u_i, v_i, t_i, y_i)\}_{i=1}^{i=N}$, where $y_i$ is a marker at time $t_i$ and usually is a discrete variable. In practice, the markers have different meanings in distinct scenes. A marker can be treated as a magnitude in modeling earthquakes and aftershocks. For financial transactions, a marker can be used to label whether a transaction is a fraudulent trading or not. The joint conditional density of an interaction $(u_i, v_i, t_i)$ with marker $y_i$ is given as $P^{u_i,v_i}\left(t_i, y_i | \hat{H}_{t_i}^{u_i,v_i}\right)$. By applying the Bayesian rule , the joint conditional density can be written as: $P^{u_i,v_i}\left(t_i, y_i | \hat{H}_{t_i}^{u_i,v_i}\right) = P^{u_i,v_i}(t_i | \hat{H}_{t_i}^{u_i,v_i}) P\left(y_i | t_i, \hat{H}_{t_i}^{u_i,v_i}\right)$, where $P^{u_i,v_i}(t_i | \hat{H}_{t_i}^{u_i,v_i})$ has the same meaning as given in subsection 3.5.1, while $P(y_i | t_i, \hat{H}_{t_i}^{u_i,v_i})$ means the distribution of $y_i$ given the interaction happened at $t_i$ with interaction history $\hat{H}_{t_i}^{u_i,v_i}$. It should be noted that the history $\hat{H}_{t_i}^{u_i,v_i}$ contains the information of history markers and one can design a marker-specific intensity function like Mei & Eisner (2017b). For simplicity, in our marked temporal point processes, we assume that $P^{u_i,v_i}\left(t_i, y_i | \hat{H}_{t_i}^{u_i,v_i}\right)$ is independent of historical markers. We model $P\left(y_i | t_i, \hat{H}_{t_i}^{u_i,v_i}\right)$ as

$$P\left(y_i | \boldsymbol{h}_{u_i,v_i,t_i}\right) = \frac{\exp\left(\boldsymbol{V}_{y_i} \boldsymbol{h}_{u_i,v_i,t_i}\right)}{\sum_{y_i} \exp\left(\boldsymbol{V}_{y_i} \boldsymbol{h}_{u_i,v_i,t_i}\right)} \tag{9}$$

where $\boldsymbol{h}_{u_i,v_i,t_i}$ is the concatenation of $\mathbf{h}_{u(t_i)}$ and $\mathbf{h}_{v(t_i)}$ which can be regarded as a dynamic representation for an interaction between $u$ and $v$ at $t_i$, $\boldsymbol{V}_{y_i}$ is the weight parameters for the $y_i th$ class. Then the overall cost function is $\mathcal{L}_2 = \mathcal{L}_1 + \mathcal{L}_{cross-entropy}$, where $\mathcal{L}_{cross-entropy}$ is a cross-entropy loss over marks:

$$\mathcal{L}_{cross-entropy} = -\sum_{i=1}^{N} \sum_{y_i} y_i \cdot \log P\left(y_i | \boldsymbol{h}_{u_i,v_i,t_i}\right) \tag{10}$$

## 4 RELATED WORK

Inspired by the Skip-gram (Mikolov et al., 2013) for word embedding, a series of node embedding methods based on the random walks on graphs have been proposed(Perozzi et al., 2014; Tang et al., 2015; Grover & Leskovec, 2016; Wang et al., 2016; 2017). GCN and its variants (Bruna et al., 2013; Hamilton et al., 2017a; Kipf & Welling, 2017) are a recent class of algorithms which extend convolutions from spatial domains to graph-structured domains. Meanwhile they can efficiently generate node embeddings for previously unseen data. All models above are designed for static graphs. The intuitive and popular approaches for modeling dynamic graphs are based on a sequence for graph snapshots(Goyal et al., 2018; Zhou et al., 2018; Seo et al., 2018; Yu et al., 2018), but it can be difficult to specify the appropriate aggregation granularity. Nguyen et al. (2018) adds a temporal constraint on random walk sampling, but it can't model the rich temporal information explicitly. Temporal point processes (TPPs) are an another alternative to model dynamics(Daley & Vere-Jones, 2007). Several dynamic graph modeling methods based on the TPPs (Dai et al., 2016; Trivedi et al., 2019) have been proposed. Our method DIP differs from these TPP-based methods by the extension of the LSTM model over temporal dependency graphs, the multiple time resolution modeling via stacking, fusing and the selection mechanism. A recently proposed method Kumar et al. (2019)

models interactions directly by predicting the next interaction embedding. More detailed related work are included in **Appendix.**C.

## 5 EXPERIMENTS

We evaluate the proposed DIP model for **interaction prediction** and **interaction classification** on several real-world datasets.

### 5.1 BASELINES AND EVALUATION METRICS

**GraphSage**(Hamilton et al., 2017a) is an inductive graph neural network framework consisting of three different aggregators which are **GCN**, **Mean** and **LSTM** aggregators respectively. We report the best results among these three aggregators noted as Graphsage*. What's more, for comparing with **GAT**(Veličković et al., 2017) we also implement a graph attention aggregator based on GraphSage. **CTDNE**(Nguyen et al., 2018) is a newly-proposed temporal network embedding method and also a tranductive method like DeepWalk(Perozzi et al., 2014). It incorporates temporal order constraint when sampling walks from time-continuous graphs. **DynGEM**(Goyal et al., 2018) takes a sequence of static graph snapshots as inputs to learn node embeddings by a deep auto-encoder network. **DeepCoevolve** (Dai et al., 2016) models dynamic interaction sequences with two co-evolution recurrent neural networks. Hidden embeddings are learned for interactive nodes after each interaction. **DyREP** (Trivedi et al., 2019) uses a two-time scale deep temporal point process model to capture dynamics of graphs. **JODIE** (Kumar et al., 2019) models interaction processes in a novel way by predicting the next interaction embedding directly instead of modeling the intensity function. For interaction prediction, we report **Mean Rank** results. To evaluate the effectiveness of top-n, we also report performances on **hit@1** and **hit@5**. As for interaction classification, we employ **KS** (Kolmogorov, 1933) score as well as **AUC**(Area under the ROC Curve) score.

### 5.2 EXPERIMENTAL SETTING

We conduct all the experiments with a hyper-parameter grid search strategy. For all methods, we search the dimension of embedding from {16, 32, 64, 128} and the learning rate from {0.01, 0.001, 0.0005, 0.0001, 0.00001}. For our DIP model, we go through {1, 2, 3, 4} for $k$ and {1, 2, 3} for $L$. For Graphsage, the maximum number of 1-hop and 2-hop neighbor nodes are set to 25 and 20 respectively. All the models are trained for at most 50 epochs with an early-stop operation if the performance does not improve for 5 epochs. For Graphsage, DynGEM ,DeepCoevolve and JODIE, we use the open source codes provided by the authors. We implement the CTNDE and GAT based on the Graphsage framework, and implement DyREP based on the pytorch implementation of DeepCoevolve. After the best configuration is found, we repeat the full experiments 5 times and report the mean results and standard deviation.

### 5.3 INTERACTION PREDICTION

#### 5.3.1 DATASETS

**CollegeMsg**(Leskovec & Krevl, 2014) consists of sending message interactions on an online social network at the University of California, Irvine during 193 days. **Ubuntu**(Leskovec & Krevl, 2014) is a temporal interaction dataset extracted from the stack exchange website. An interaction between two users means one answered another's questions or replied to his/her posts. **Amazon**(McAuley et al., 2015) is composed of commodity rating data from amazon users. We use the **Clothing** subset of this dataset. **MathOverflow**(Leskovec & Krevl, 2014) is comprised of interactions of commenting an existing answer on the *Math Overflow* website. **Table 1** shows the detailed dataset statistics. In this table, **Repetition** is the rate of repeated interaction in datasets. For each dataset,we first sort these interactions by occurrence time and split them to be training/validation/test sets. The cold-start participants which only exist in validation set or test set are removed.

Table 1: Dataset Statistics for interaction prediction

|  | CollegeMsg | Ubuntu | Amazon-Clothing | Math Overflow |
|---|---|---|---|---|
| # Train/ # Valid/ # Test | 35902/7814/5055 | 204846/39913/35271 | 50209/9195/7598 | 58596/24045/32705 |
| Repetition in Valid(%) | 72.55 | 56.60 | 0.00 | 67.42 |
| Repetition in Test(%) | 75.67 | 67.57 | 0.00 | 79.92 |
| Duration(days) | 193.63 | 2587.96 | 3657.00 | 2350.12 |

### 5.3.2 RESULTS AND ANALYSIS

Figure 4 summarizes the **Mean Rank** performances of all methods. On the whole, our DIP method consistently beats all baselines by 65.84%, 41.64%, 10.69% and 43.99% over the four diverse datasets. The **CTDNE** method performs worst across all the datasets since the generated embedding is static and can't be updated and evolved across validation and test datasets. Meanwhile we can see that there is no consistent winner among baselines and all the methods perform relatively better on the **CollegeMsg**, **Ubuntu** and **MathOverflow** datasets than do on the **Amazon-Clothing** dataset. It is also noteworthy that the static methods **GAT** and **GraphSage\*** perform competitive with dynamic baselines on these three datasets. These phenomenons above could be explained that the **CollegeMsg**, **Ubuntu** and **MathOverflow** have lots of repetitive interactions (at least 75% for **CollegeMsg** and **MathOverflow** on test data, 68% for **ubuntu** on test data as shown in Table 1) and repetitive information makes recurring interaction predicted easily. The two static methods perform worse than the other dynamic methods(but **JODIE** ) on the **Amazon-Clothing** because no repetitive information can be reused and nodes representation can't be updated across validation and test like **CTDNE**. As for comparison with dynamic methods, our work performs better than the **Dyrep**, **DeepCoevolve** and **JODIE** which use the similar mutually-recursive RNNs to capture and update co-evolution states sequentially. We argue that the vallina RNNs can't capture long-term dependency history well and have optimization problems which may lead to worse performances than ours. Moreover they treat past history information equally while our selection mechanism can select more relevant information for updating dynamic representation. The performances of all methods on hit@1 and hit@5 are given in **Appendix.B**. The effects of k and L are also investigated in **Appendix.B**.

## 5.4 INTERACTION CLASSIFICATION

### 5.4.1 DATASETS

We conduct this task on an industrial dataset: **Huabei Trade Data**. This dataset consists of about 150,000 transaction records processed by Huabei during August 2018. Each transaction is initiated with three parties: the buyer, the seller and transaction details such as merchant category and transaction amount. Around 15% of the transaction are fraudulent and is labeled by a complicated *Ex-Post* method. For each interaction event, there are 11 context features including information about buyer types, seller types, purchased items' categories and trading platform. We use the first 10 days data as training set, the following 10 days data as validation set, the rest as test set. Note that, in this scenario, there are users who only appear in validation/testing dataset. Thus, the *transductive* method **CTDNE** is not applicable on this task. Meanwhile, since the dynamic baseline methods are all unsupervised, we only report GCN and our marked DIP methods Alternatively, we employ the **XGBoost** (Chen & Guestrin, 2016) as an additional baseline which is a popular baseline method in the cash-out detection task(Hu et al., 2019). The detailed data statistics are given in **Appendix.B.1**.

### 5.4.2 RESULTS AND ANALYSIS

Table 2 compares the results. Obviously, our model outperforms all the baseline methods with large margins. The **Xgboost** method which only utilize interaction context feature can't model any co-evolution information in time-evolving interactions, thus it performs worst. The GCN-related methods perform worse than our DIP since the static construction of interaction graph can blur temporal structural information and miss time-interval information. A toy example in figure.1 can explain why static graph methods fail in this task.

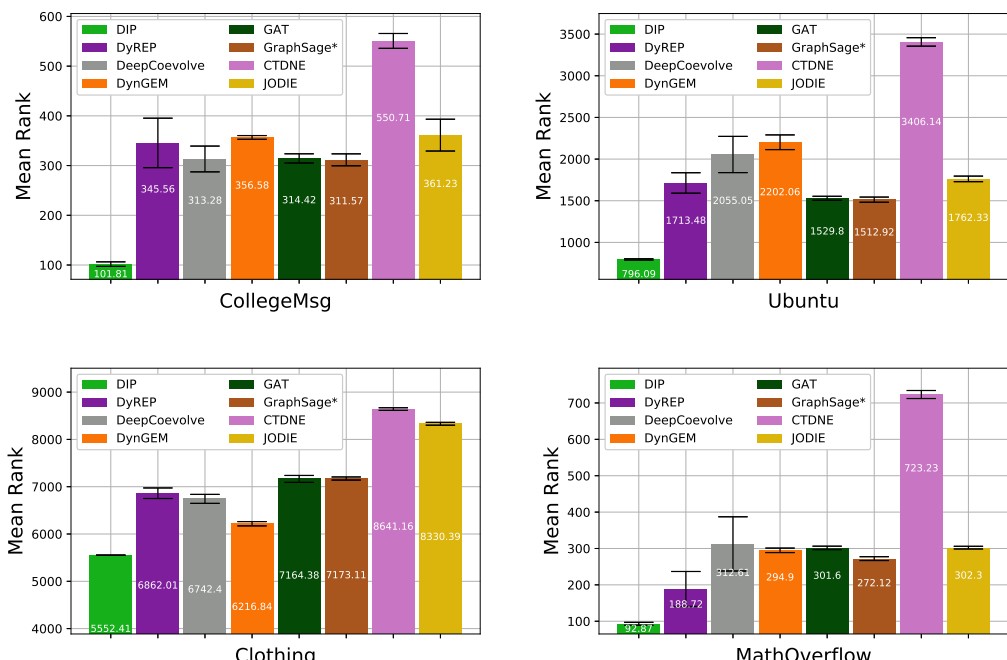

Figure 4: Mean rank results. As low mean rank indicates that the ground-truth item is ranked accurately, we can observe that DIP always outperforms baselines.

Table 2: Interaction classification results

|     | Xgboost | GraphSage* | GAT | DIP |
|-----|---------|------------|-----|-----|
| AUC | 0.6818 ±0.0023 | 0.8603 ±0.0005 | 0.8597 ±0.0004 | **0.9017** ±0.0004 |
| KS  | 0.2536 ±0.0015 | 0.5934 ±0.0012 | 0.6018 ±0.0005 | **0.6703** ±0.0060 |

## 5.5 ABLATION STUDY

As we described in Section 3, the DIP model consists of three important components: First, it uses a Time Gate in the DIP neural unit to explicitly model the temporal information. Second, the selection mechanism enables our model to select more important historical information for interactions. Third, the Fusion of multi-layer DIP-UNIT's hidden state vector helps to extract high level feature. We investigate the contribution of each component by disabling each of them one by one, and compare the corresponding result to the full model. figure 5 and figure 6 give the detailed ablation results. **FullModel** in the two figures means all the three components are enabled.

- **No Time Gate:** In this configuration, the time gate in DIP-UNIT is disabled. This leads to a significant drop of the Mean Rank performance. It provides a strong evidence for the effectiveness of the time gate.
- **No Selection:** In this configuration the selection mechanism is disabled. Accordingly, all the historical node representations contribute equally, thus again leading to a performance drop.
- **No Fusion:** In this variant, we directly use the hidden state vector of the last layer. Again, the performance degrades significantly. This demonstrates that a fusion of different layers' representations gives richer information than the last layer only.

## 6 CONCLUSIONS

In this paper, we have proposed a deep multidimensional point process approach, DIP, to learn dynamic graph representations. We generalize LSTM over temporal dependency graphs and model

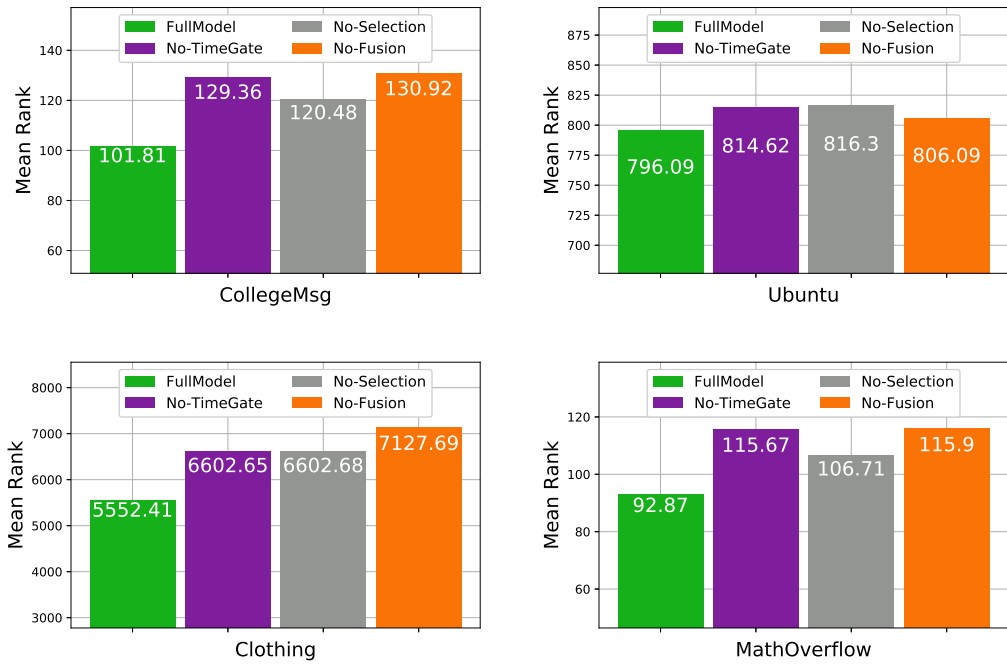

Figure 5: Ablation study results of the interaction prediction task. Disabling any one of the three component leads to a performance drop.

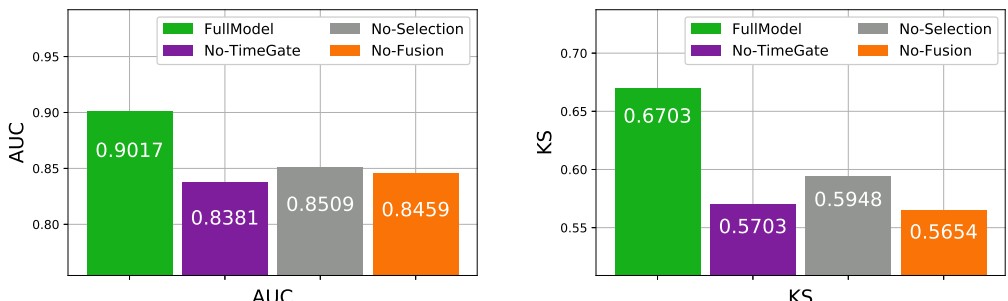

Figure 6: Ablation study results of the interaction classification task. All of the three component contributes a lot to improve the final classification precision.

multiple time resolutions via stacking, selection and fusion. Experimental results show the effectiveness of the components of our neural unit and the superior performance on several datasets.

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

## A  TEMPORAL DEPENDENCY GRAPH

The key step for efficient training or inference is to fast obtain *subgraph*$(u(t), k)$ or *subgraph*$(v(t), k)$ for an interaction $(u, v, t)$. Obviously, our definition of temporal dependency graph(*TDG*) in Subsection.3 provides a recursive and incremental way for its construction. As illustrated in figure 2, we can easily find that *subgraph*$(u(t), k)$ is the union of *subgraph*$(u(t), k)$, *subgraph*$(w(t^-, k))$ and the newly added dependency edges. Based on this feature, to fast obtain $k-$depth *subgraph*, we design two algorithms, TDG-COLORING and CONSTRUCTION OF K-DEPTH TDG orderly.

### A.1  TDG-COLORING

The TDG-COLORING algorithm is shown in Algorithm 1, which is the pre-step of Algorithm 2. This algorithm takes $I = \{(u_i, v_i, t_i)\}_{i=1}^{i=N}$ as input and generate a new sorted sequence of interactions ordered by color numbers in an ascending way. The TDG-COLORING algorithm ensures that, interactions in the same color group are independent and interactions from groups with smaller color numbers are precedents of the larger ones.

---

**Algorithm 1** TDG-COLORING

---

**Require:** $I$: A sequence of interaction with a chronological order.
**Ensure:** $ColorGpSeq$: A sequence of sorted interactions by color no.
1: Initialize $ColorGroupArray[x] \leftarrow -1$ ▷ x represents nodes in $I$ and assign an initial color no -1 for all nodes
2: Initialize $LastNodeTime[x] \leftarrow -1$  ▷ Record the latest time when node x was involved in an interaction and initialize with -1
3: **for** $event$ in $I$ **do**
4:    $cur\_u, cur\_v, cur\_time = event.u, event.v, event.t$
5:    **if** $cur\_time > LastNodeTime[cur\_u]$ **then**
6:       $ColorGroupArray[cur\_u] + +$
7:       $LastNodeTime[cur\_u] = cur\_time$
8:    **end if**
9:    **if** $cur\_time > LastNodeTime[cur\_v]$ **then**
10:       $ColorGroupArray[cur\_v] + +$
11:       $LastNodeTime[cur\_v] = cur\_time$
12:    **end if**
13:    $event.gn = \max(ColorGroupArray[cur\_u], ColorGroupArray[cur\_v])$  ▷ gn is short for group no
14: **end for**
15: $ColorGpSeq = Sort(I)$       ▷ sort $I$ by an ascending order with assigned group color no.
16: **return** $ColorGpSeq$

---

### A.2  CONSTRUCTION OF K-HOP TDG

The details of CONSTRUCTION OF K-DEPTH TDG are given in Algorithm 2. In Algorithm 2, based on the definition of TDG and $ColorGpSeq$ output by Algorithm 1, we can incrementally construct the k-depth subgraphs for any two nodes in a new interaction. Specifically, we first call PREVIOUS function to get each node's adjacent interaction in which it was involved before current interaction. Then incrementally construct a graph($ugraph$) rooted at $cur\_u$(lines 6 to 10). Here $H_{k\_subgraph}[u\_preInteraction]$ stores the $subgraph$ of $ugraph$ which was obtained before $ugraph$ because $u\_preInteraction$ ranks ahead of current $event$ in $ColorGpSeq$. Likely, we can get $vgraph$ incrementally. At last, we call the Breadth-First-Search algorithm with the traverse $depth = K$ to obtain the k-depth subgraphs for $cur\_event$ and then store it into $H_{k\_subgraph}[cur\_event]$.

### A.3  OBTAINING AND UPDATING K-DEPTH TDG

As for efficiency, fast obtaining k-depth subgraph of TDG for an interaction is both important for offline training and online inference. Based on Algorithm 2, for an interaction $l$ from the

---

**Algorithm 2** CONSTRUCTION OF K-DEPTH TDG

---

**Require:** $ColorGpSeq$: A seq of sorted interaction by group color no.
**Ensure:** HashTable $H_{k\_subgraph}$: Map an interaction to its corresponding $k\_subgraph$.
  1: Initialize an empty HashTable $H_{k\_subgraph}$
  2: **for** $cur\_event$ in $ColorGpSeq$ **do**
  3:     $cur\_u, cur\_v = cur\_event.u, cur\_event.v$
  4:     $u\_preInteraction = Previous[cur\_u]$ ▷ *Previous* Function returns last interaction in which $cur\_u$ was involved
  5:     $v\_preInteraction = Previous[cur\_v]$
  6:     **if** $u\_preInteraction$ exits **then**
  7:         $edge_1 = (u\_preInteraction.u, cur\_u)$            ▷ The edge is defined according to the definition of dependency graph
  8:         $edge_2 = (u\_preInteraction.v, cur\_u)$
  9:         $ugraph = edge_1 \cup edge_2$
 10:         $ugraph = H_{k\_subgraph}[u\_preInteraction] \cup ugraph$            ▷ Incremental Update
 11:     **else**
 12:         $ugraph = cur\_u$
 13:     **end if**
 14:     **if** $v\_preInteraction$ exits **then**
 15:         $edge_3 = (v\_preInteraction.u, cur\_v)$
 16:         $edge_4 = (v\_preInteraction.v, cur\_v)$
 17:         $vgraph = edge_3 \cup edge_4$
 18:         $vgraph = H_{k\_subgraph}[v\_preInteraction] \cup vgraph$            ▷ Incremental Update
 19:     **else**
 20:         $vgraph = cur\_v$
 21:     **end if**
 22:     $G_{K-depth} = BFS(ugraph \cup vgraph, depth = K)$ ▷ Call Breadth-First-Search with max $depth = K$
 23:     $H_{k\_subgraph}[cur\_event] = G_{K-depth}$
 24: **end for**
 25: **return** $H_{k\_subgraph}$

---

collected training interaction data, we can obtain its corresponding k-depth subgraphs directly using $H_{k\_subgraph}[l]$ with time complexity $O(1)$. For a new incoming interaction $l$ with two nodes $cur\_u$ and $cur\_v$, we can easily reuse Algorithm 2's code from line 3 to 22 to find $ugraph$ and $vgraph$, respectively. Then we merge them to get k-depth subgraph for doing inference online. At the same time, we can incrementally update TDG by $H_{k\_subgraph}[l]$. The time complexity here for updating *TDG* mainly depends on union between $ugraph$ and $vgraph$ in line 22 of Algorithm 2 and is $O(n+e)$ where $n$ and $e$ are the total number of nodes and edges for $ugraph$ and $vgraph$.

# B   ADDITIONAL DATA STATISTICS AND EXPERIMENT RESULTS

## B.1   HUABEI TRADE DATASET

Table 3: Huabei Dataset Statistics

|       | # Interaction | # Seller | # Buyer | Repetition(%) |
|-------|---------------|----------|---------|---------------|
| Train | 60705         | 19453    | 22916   | -             |
| Valid | 53609         | 16669    | 23517   | 25.21         |
| Test  | 34471         | 9929     | 17549   | 23.55         |

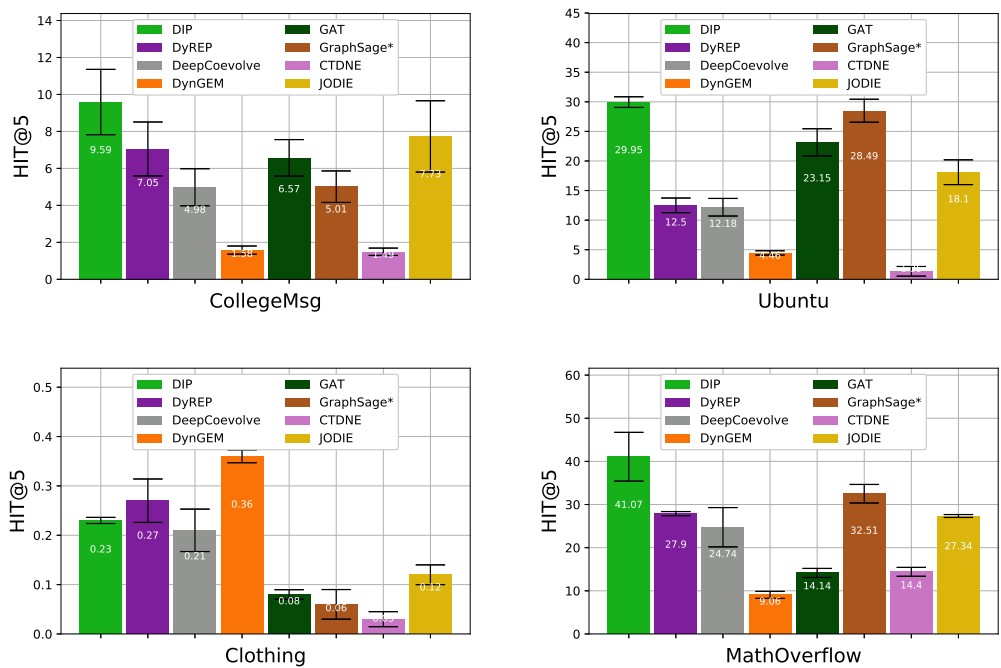

Figure 7: Hit@5 Results

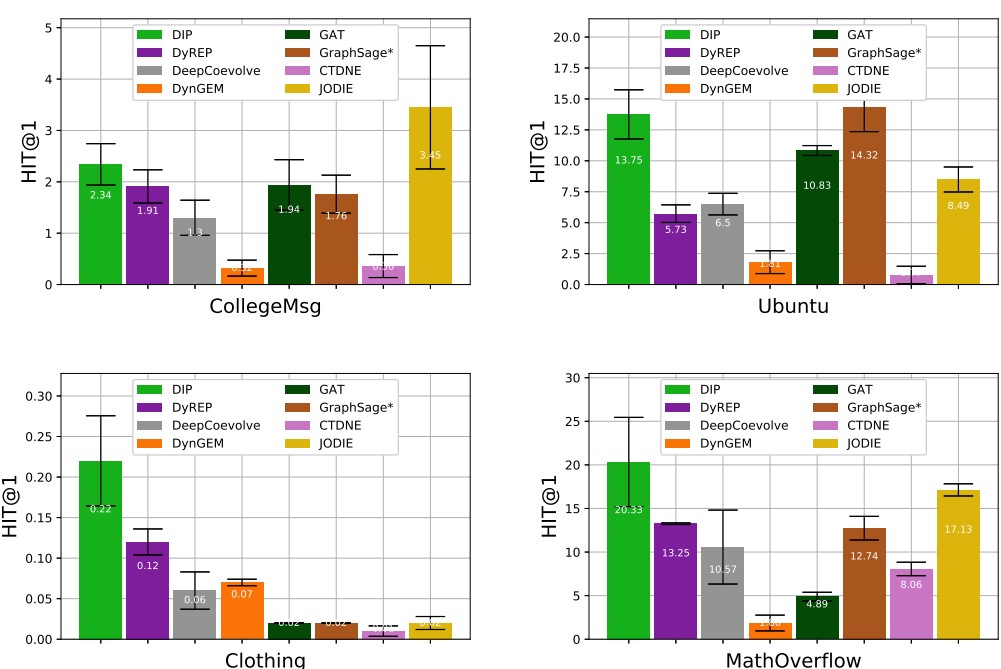

Figure 8: HIT@1 Results

## B.2 HIT PERFORMANCE

Figure 7 and Figure 8 provide HIT@5 and Hit@1 results in addition to the Mean Rank results in Section 5.3.2 of the main paper. HIT@n is defined as $HIT@n = \frac{\sum \delta_i}{\#of\ Test\ Interaction}$, where $\delta_i = 1$ if $rank_i <= n$ else 0, which measures the ability of top rank prediction.

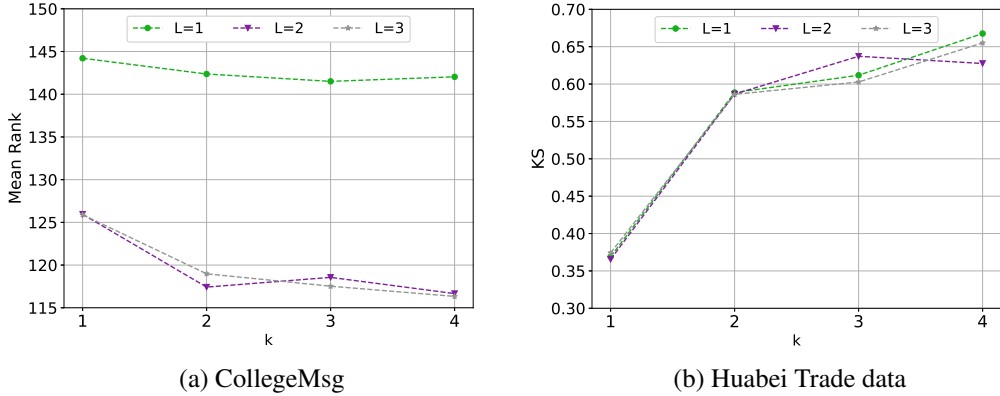

(a) CollegeMsg              (b) Huabei Trade data

Figure 9: Sensitivity to k and L.

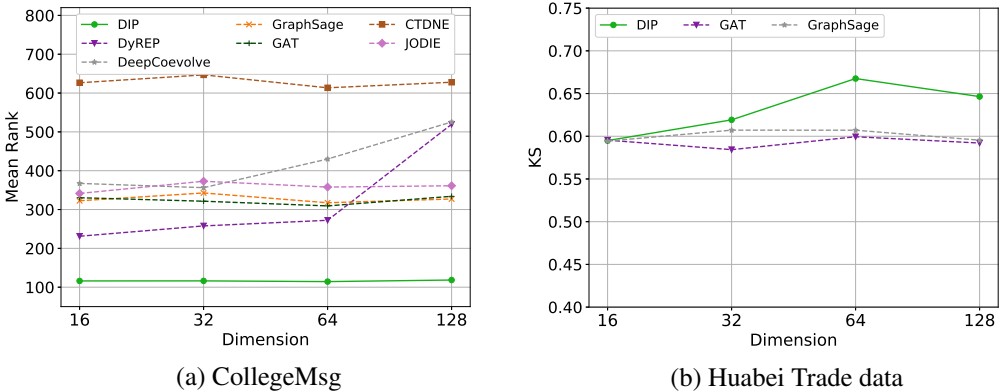

(a) CollegeMsg              (b) Huabei Trade data

Figure 10: Sensitivity to dynamic representation dimension.

### B.3 EFFECT OF THE DEPTH K AND THE NUMBER OF LAYERS L

For a given node $u(t)$, only the history information in k-depth temporal $subgraph$, i.e, $subgraph(u(t), k)$, is considered to calculate the dynamic representation, so the number of depth $k$ could affect the final representation. Meanwhile, our proposed fusion mechanism is based on deep model which can capture different information at different layers. So, in this subsection, we investigate the effects of $k$ and $L$ for CollegeMsg dataset on the Interaction Prediction task and Huabei Trade dataset on the Interaction Recognition, which is shown in Figure 9. As we can see, stacking multiple layers does help improve the interaction prediction and interaction classification performances, but it doesn't mean the larger L will always give the better results. And with the k increases which means we could utilize more history information, the trend of the performances go better for both tasks.

### B.4 EFFECT OF EMBEDDING SIZES

To investigate the effect of the dynamic embedding size on the interaction prediction and interaction classification tasks, we vary the dynamic embedding dimension in {16, 32, 64, 128} and report the corresponding Mean Rank results for interaction prediction on the CollegeMsg dataset and KS results(AUC is similar) for interaction classification on Huabei Trade Data in figure 10. In figure 10 (a), we see that the dynamic embedding size has little impact on DIP model, and it consistently outperforms all baselines. The performances of several baselines drop more or less when the embedding size is set to 128. The baseline model could overfit at this embedding size. In figure 10 (b) we find that when all the baselines are set with the same dynamic embedding sizes, our DIP model always outperforms the competitors.

## C    DETAILED RELATED WORK

Graph representation learning, which is also known as graph embedding or network embedding, is a task aiming to learn low-dimensional dense vectors for vertex and edges that preserve the original graph structural information and network properties. Before the era of deep learning, conventional graph representation learning methods usually adopts dimension reduction (Belkin & Niyogi, 2002; Tenenbaum et al., 2000; Roweis & Saul, 2000) or matrix factorization(Ahmed et al., 2013) techniques. While these methods usually suffer a heavy computational cost. After the great success of deep learning methods in the field of computer vision and natural language processing, especially Skip-gram(Mikolov et al., 2013) in word representation learning, this problem has been widely investigated by neural network methods. We review the development of this research direction from the following perspectives.

**Static Graph Representation Learning**: Perozzi et al. (2014) proposed the DeepWalk model utilizing random walks on graph to generate sequences of nodes, and feed them into the Skip-gram model to learn nodes' low dimensional representation vector. Then, Tang et al. (2015) proposed the definition of first-order proximity and second-order proximity in the model named LINE, that jointly models this two different level structure information. Grover & Leskovec (2016) designed a biased random-walk sampling method to capture more flexible neighborhood information which is helpful to learn richer representations. And Wang et al. (2016; 2017) extended these models by considering high-order proximity and community structure. But these mentioned methods are somehow shallow. Another type of method is the Graph Convolution Network (Bruna et al., 2013; Kipf & Welling, 2017). They extended the convolution neural network to the graph spectral space, thus applied DEEP model on graph data. While these methods are still *transductive*, and can not jointly models the network structure information and the attributed on nodes or edges of graph. To overcome this problem, Hamilton et al. (2017a) proposed an inductive graph embedding framework, which divided the representation learning into two phrase: sampling and aggregation, such that it could incorporates node feature information and thus generate embedding from these features using the well-trained graph neural network for unseen nodes. Veličković et al. (2017) used self-attention to learn weights for neighborhood, then aggregated them by the self-adapted weight. While, most of these approaches can only model static graph data.

**Dynamic Graph Representation Learning:** A popular approach for modeling dynamic graph data is considering the dynamics as a sequence of graph snapshot(Soundarajan et al., 2016). Zhu et al. (2016) uses an non-negative matrix factorization technique to embed social network in a temporal latent space. Zhou et al. (2018) models the specific triadics closure formation procedure over the snapshots. Goyal et al. (2018) adapts an graph autoencoder and learns a stable embedding over time. Seo et al. (2018) combines a CNN module and a RNN module to capture spatial characteristics and temporal characteristics. NetWalk(Yu et al., 2018) also learns vertex representations from sequences of snapshots, and it is designed specially for anomaly detection. In contrast, Nguyen et al. (2018) adds temporal order constraint on random walk sampling to capture evolving neighborhood. But, it can't explicitly models the rich temporal information.

**Temporal Point Process:** Temporal Point Process is a powerful statistical tool for modeling sequences of events with unequal interval. It has been wildly used in recent research with different intensity function: from parametric(Du et al., 2016) to recurrent neural networks(Dai et al., 2016; Mei & Eisner, 2017b), event reinforcement learning based(Xiao et al., 2017). DeepCoevolve(Dai et al., 2016) designs a recurrent neural network to capture the co-evolution dynamics of interaction data. But it just takes the interaction counterpart into the co-evolution module, which may loss temporal structural information. DyREP(Trivedi et al., 2019) divides the interactions between nodes into two different types as *communication* and *association* and models them separately. Although it uses the neighborhood of interaction counterpart to update node representation, the relevance between the different neighbors of counterpart and the node itself has not been modeled.

