# OpenReview forum: "Deep Interaction Processes for Time-Evolving Graphs"
_ICLR.cc/2020/Conference — Reject_

### Official Review · AnonReviewer1 · 2019-10-25
**Official Blind Review #1**

**Rating:** 3

**Review:**

This paper considers modeling continuous time-evolving graphs using a temporal point process framework. It introduces a time gate in the LSTM to handle the temporal dependency and uses an attention mechanism to select relevant nodes to learn the underlying dynamics.

Overall, this paper is not easy to understand in detail. Firstly, it is unclear how and why the temporal point process can deal with growing/shrinking graph nodes and changing interactions. Secondly, how does the DIP-UNIT handle the continuous graph changing? What if the graph changes with an uneven speed? Thirdly, how do all the small pieces work together to achieve the goal of the paper? An overview diagram or a toy example would greatly improve the readability of the paper.

Besides, what is the computational cost of the proposed network? How large a graph could be and how fast its changes could be captured?

**Experience Assessment:**

I do not know much about this area.

**Review Assessment: Checking Correctness Of Derivations And Theory:**

I did not assess the derivations or theory.

**Review Assessment: Checking Correctness Of Experiments:**

I did not assess the experiments.

**Review Assessment: Thoroughness In Paper Reading:**

I read the paper at least twice and used my best judgement in assessing the paper.

---

> ### Author Response · Authors · 2019-11-14
> **Response to Review #1**
>
> Thanks for your review and valuable advice, we have added an overview diagram as the illustration of the whole pipeline of our method. Please see the new Figure 2 for details.
>
> Temporal Point Process is a powerful mathematical tool for modeling sequences of interactions[1]. The ability to discover correlations among interactions is crucial to accurately predict the future of a sequence given its past, i.e., what interactions are likely to happen next, when they will happen and between which participants. And the key point to characterize temporal point processes is via the conditional intensity functions λ(t). Formally, λ(t)dt is the conditional probability of observing an event in a small window [t, t + dt) given the history H(t) up to t and that the event has not happen before t.
>
> In this paper, we adopt a neural approach named DIP to model the conditional intensity functions given all past graph-structured dependency history.
>
> As the toy-example shown in Figure 2 , The pipeline of our work is as follows :
> 1. Propose a temporal dependency graph(TDG) concept to depict complex dependencies among time-evolving interactions.
> 2. Learn representation using DIP units, i.e. ,graph-structured LSTM with time gates to handle irregular time intervals.
> 3. Considering computational burden, when an interaction(event) happened, we only use its k-depth subgraph to compute their new representations(Fig2.d) which is similar to chain-LSTM training unfolded with max k steps and k-hops setting in static graph.
> The figure 3 gives an illustrative example for Selection and Fusion mechanism.
>
> As for your question
> 1. It is unclear how and why the temporal point process can deal with growing/shrinking graph nodes and changing interactions ?
> ANS1: The above description show the whole pipelines.
>
> 2. how does the DIP-UNIT handle the continuous graph changing?
> ANS2: Each time the graph changes, which means there are new interactions. According to Algorithm 2 in Appendix A, we can incrementally update the temporal dependency graph. Then,  we feed the new temporal subgraph of the involved nodes into the neural network, and update the dynamic representations for them.
>
> 3. What if the graph changes with an uneven speed?
> ANS3: The design of the time gates is intended to handle incoming interaction with different time intervals(i.e., uneven speed). Specifically, the time gates combined with forget gates control the information flow in and out to the  memory cells of the DIP units.  Then the final representation and intensity function not only have graph-structure information, but also frequencies and speed of an evolving graph.
>
> 4. How large a graph could be  ?
> ANS4: In this paper, we focus on time-evolving interactions, so the graph is growing larger and larger with time as long as a new interaction occurs. As for efficiency, Algorithm2 in Appendix A provides a way to construct k-depth subgraph incrementally with o(m+n) time complexity where m, n are the nodes in subgraphs. Meanwhile, there is a big advantage we can train our model in parallel since we only consider k-depth subgraphs while the baseline methods can only train and update the states sequentially.
>
> 5. how fast its changes could be captured?
> ANS5: we are processing continues time-evolving graph. See the answer of question 2.
>
>
> [1]DJ Daley and D Vere-Jones. An Introduction to the Theory of Point Processes: Volume I: Elementary Theory and Methods. 2007.

---

### Official Review · AnonReviewer2 · 2019-11-03
**Official Blind Review #2**

**Rating:** 3

**Review:**

The paper is concerned with modeling continuous time-evolving graphs, for which it proposes to combine temporal point processes with a recurrent architecture to learn dynamic node representations. In addition, the paper proposes to stack multiple recurrent layers (to obtain node representations over multiple time scales) and use a temporal attention mechanism (to select relevant past interactions).

Modeling temporal and dynamic graphs is an important problem with many applications in ML and AI. The focus of the paper, i.e., to develop improved models by combining TPPs and representation learning, is a promising approach to this task and fits well into ICLR. Furthermore, the presented experimental results are promising.

However, I'm concerned about different aspects of the current version: The main contributions of the paper are a recurrent (LSTM-based) architecture to model the intensity function of a TPP, stacking multiple LSTM to form a deep architecture, and a temporal attention mechanism. However, none of these contributions on its own are particularly novel. For instance, prior work that introduces similar approaches include
- Recurrent networks to parameterize intensity functions: (Dai, 2017), (Mei & Eisner, 2017), (Trivedi, 2019), ...
- Temporal attention: (Trivedi, 2019)

Hence, the main novelty seems to lie in the stacked architecture and the particular combination of modules (which is of limited novelty). The experimental results are certainly interesting, but it would be important to provide a more detailed analysis of the model to get insights into the causes for these improvements.

With regard to the model: The Log-likelihood function in Section 3.6.1 seems to be incorrect as the LL for a TPP would be L = \sum_{i:t_i \leq T} \log\lambda(t_i) - \int_0^T \lambda(s)ds, which is quite different from the equations in the paper. Is the LL in Section 3.6.1 the actual objective that has been optimized?

Regarding the experimental results: All models are trained on the same grid of embedding dimensions, but the proposed method is the only deep model. Hence, its maximum number of parameters can be up to 4x compared to the shallow models. How do the results look if all for models with comparable number of parameters (i.e., can the improvements be explained due to this difference)? It would also be good to get results on commonly used benchmarks (e.g. data used in DyRep or NeuralHawkes) to make the results of the new model comparable to prior experiments and datasets.

**Experience Assessment:**

I have read many papers in this area.

**Review Assessment: Checking Correctness Of Derivations And Theory:**

I assessed the sensibility of the derivations and theory.

**Review Assessment: Checking Correctness Of Experiments:**

I assessed the sensibility of the experiments.

**Review Assessment: Thoroughness In Paper Reading:**

I read the paper at least twice and used my best judgement in assessing the paper.

---

> ### Author Response · Authors · 2019-11-14
> **Response to Review 2**
>
> Thank you for your detailed comments and suggestions. We have already updated our paper with a more clear toy example as shown in Fig.2. and Fig.3.  Please see it and maybe  help u understand details of our work better.
> Now I will answer your Questions as follows:
>
> Q1: However, I'm concerned about different aspects of the current version: The main contributions of the paper are a recurrent (LSTM-based) architecture to model the intensity function of a TPP, stacking multiple LSTM to form a deep architecture, and a temporal attention mechanism. However, none of these contributions on its own are particularly novel. For instance, prior work that introduces similar approaches include Recurrent networks to parameterize intensity functions: (Dai,2017),(Mei\&Eisner,2017), (Trivedi, 2019), - Temporal attention: (Trivedi, 2019)
>
> A1:  The differences between our method and Mei \& Eisner, 2017 are as follows: First, our work focus on time-evolving interaction events which is based on a multi-dimension point process with each
> user-item pair as one dimension while their work are mainly based on a one-dimensional point process and they only consider chain-structure dependencies among events without considering interaction.  Second, They view their intensity function as a nonlinear function of chain-structure histories  while our intensity function is a nonlinear function of representation based on complex graph-structured histories using DIP units.
>        As for the work in Dai2017, they use mutually-recursive RNNs and  incorporate the participants’ embedding to capture the dynamics of coevolution. To capture co-evolution in time-evolving graph, we propose the selection Mechanism which is used to weight important interaction in their k-hop history subgraphs of current interaction. The weighting operation is based on mutual information among interactive k-depth subgraphs. In addition, simple RNN can't capture long history well.
>        As for the temporal attention method in Trivedi 2019,  we have two attention operations applied in this paper but they have different purposes.  The first one is a co-attention operation in the Selection Mechanism ( see Section 3.3.2 and Fig.3 ) .  The Selection Mechanism is intended to select and weight important interaction in their k-hop history subgraphs of current interaction. Specifically, a co-attention operation is first used to capture mutual information among two k-hop history subgraphs  of two interactive nodes. Then based on mutual information, adaptive gates are learned to weight each history node in their corresponding k-hop history subgraphs.  However, the method in Trived 2019 only consider one-hop temporal neighbors when updating nodes' dynamic representation with a simple self-attention function.
>
> Q2: With regard to the model: The Log-likelihood function in Section 3.6.1 seems to be incorrect as the LL for a TPP would be $L = \sum_{i:t_i \leq T} \log\lambda(t_i) - \int_0^T \lambda(s)ds$, which is quite different from the equations in the paper. Is the LL in Section 3.6.1 the actual objective that has been optimized?
>
> A2:  We carefully checked the equation and found that we missed the log symbol for intensity which was a writing mistake. It was updated now. Additionally, the equation you gave here is for a one-dimensional point process while our equation in Section 3.5.1 is a multi-dimensional point process whose survival function is a summation for all possible interaction pairs(i.e, multi-dimensions). (the same as in Trivedi 2019).
>
> Q3: Hence, the main novelty seems to lie in the stacked architecture and the particular combination of modules (which is of limited novelty). The experimental results are certainly interesting, but it would be important to provide a more detailed analysis of the model to get insights into the causes for these improvements.
>
> A3: It seems that there is a misunderstanding on this point. As explained above, our contributions are as follows(as shown in fig.2 and fig.3):  1. we define a temporal dependency graph($TDG$) 2. we generalize the traditional chain-structured LSTM to a graph-structured LSTM with time gates(named DIP units) to depict nodes' dynamic representation in $TDG$.  3. we enhance the nodes representation by a novel selection method using a two-phase gating operation and a fusion mechanism to integrate all layers' information. 4. the state of art methods like DeepCoevol and JODIE use a RNN-like equation to update nodes states incrementally which limits computation parallel and have efficiency problems. Moreover, simple RNN  can't capture dynamics in long sequences well, not to mention the complex interaction network with a long duration. However, our method uses a k-depth subgraph history information to update information like chain-lstm training unfolded with max k steps and max k-hops in GCN. Meanwhile the graph-lstm itself has good properties to capture long sequences well.

---

> > ### Author Response · Authors · 2019-11-14
> > **Response to Review 2-II**
> >
> > Q4: Regarding the experimental results: All models are trained on the same grid of embedding dimensions, but the proposed method is the only deep model. Hence, its maximum number of parameters can be up to 4x compared to the shallow models. How do the results look if all for models with comparable number of parameters (i.e., can the improvements be explained due to this difference)?
> >
> > A4:  Results analysis are given  in the updated sec 5.3.2 and 5.4.2 .and we have new experiments about the effects of embedding size, k and L on the two tasks--interaction classification and  interaction prediction (see Appendix B.3 B.4). Even with smaller embedding size or smaller value of k and L, our method still performs well. Meanwhile it doesn't mean larger k and L combination always give the best results. All the parameters are set according to performance at validation set.

---

### Official Review · AnonReviewer3 · 2019-11-04
**Official Blind Review #3**

**Rating:** 3

**Review:**

The paper focuses on the problem of modeling interaction processes over dynamically evolving graphs and perform inference tasks such future interaction prediction and interaction classification. Specifically, the paper proposes a temporal point process based formulation to model the interaction dynamics where the conditional intensity function is parameterized by a recurrent network. With an occurrence of any event, the recurrent architecture updates the embeddings of the nodes involved in that event which then affects the intensity function and hence the likelihood of future events. The paper uses intensity based likelihood to train for future interaction prediction task while cross-entropy based loss for classification task. The paper demonstrates the efficacy of the method through experiments across multiple datasets and compare against representative baselines and further provides ablation analysis for the proposed architecture.

The paper demonstrates markedly improved empirical performance on multiple datasets and also performs the task of interaction classification which is not seen in recent works on evolving graphs, which are plus points. However, there are several concerns with the overall work that makes this paper weaker: (1) The main concern is with the novelty and more importantly the justification/analysis of the contributions proposed approach. (2) Further, while the ablation study provides some insights into architecture, it is not adequate (3) The paper misses comparison with a very important and recently proposed baseline, JODIE [1].


Main Comments:
--------------
- The paper leverages existing techniques built for learning over evolving graphs and augments it with three modifications: explicit use LSTM with time gate, stacked LSTM approach with fusion (Aggregation) and attention mechanism to select important neighbors to contribute to embedding update. The use of LSTM with time gate and fusion mechanism is very incremental contribution.  The attention mechanism proposed here is novel compared to existing works. However, there is very little justification or analysis provide or either of the contributions. This is big drawback of this contribution.

- For instance, the authors mention that stacked LSTM is used to capture multiple resolution. Can they provide some analysis or empirical demonstration that this actually happens? Also, the authors mention they use K in range of {1,2,3,4} but do not provide details what was useful for each dataset and how is it useful. How does the scaling parameter and alpha affect the performance and what are their roles? Also, what does superscript 'task' signify?
Similarly, they propose coattention mechanism with adaptive gate functions but does not provide any analysis of why they are useful and what characteristics they capture in the data that allows it to select most relevant neighbors. Is the attention mechanism temporally dependent?

- The authors perform ablation studies by switching off each component as a whole but considering the way this architecture is built, this is not a very useful exercise except knowing that each component contributes to the performance. A more detailed analysis and ablation is required. For instance, can the authors show performance with different K and  how it deteriorates/improves with it? Also, for stacked LSTM case, the authors show what happens when you use last layer, but what happens if the authors use only one layer (I guess this is K=1?) or don't use residual connections? When the time gate is switched off, does the authors also remove deltas from intensity function? what happens in this scenario? How does subgraph depth affect the quality of performance? What happens if authors don't sue adaptive gate functions?

- Figure 2 shows an example of bipartite graph, however, it seems datasets in experiments does have non-bipartite case? Is this true or the method only works for bipartite case?

- The use of proposed Algorithm 2 is not well justified. Why does the author need coloring and hashing mechanism instead of simpler BFS/randomwalk routine to collect previous interactions? Also, is this subgraph created for each event or it is computed offline during training? Further, the subgraph used for selection mechanism same as subgraph used for backtracking in LSTM?

- Further, is it true that the training is done in order of ColorGraphSeq or is it done in order of dataset? How does the authors capture dependencies across dataset in later case?

- Do you also update cell states with selection mechanism? The DIP-UNIT equation in selection section does not show that update. Also, are the embeddings updated only during train or also during validation/evaluation?

- The authors only present the results as-is without any insights on the performance of DIP model vs others and why they are able to demonstrate good performance. It is highly desired that authors add discussion section for each set of results to provide such information

- The authors include support for new nodes for interaction classification task but remove them for interaction prediction task which is strange. Is there a specific reason for this? What is the effect on the performance if new nodes are allowed in test? Further, why is interaction classification not compared with temporal baselines? All baselines produce embeddings and the authors mention that classification for this paper is independent of marker history. While the temporal baselines do not train for the task, the authors can train a second stage classifier with learned embeddings to perform classification

- The authors do not compare with recently proposed JODIE [1] which is a big miss. The comparison is required as it also models interaction processes in a  novel way by actually predicting the next embedding directly instead of modeling the intensity. An empirical comparison and discussion of this method is required to compare with various state-of-art methods.

Minor:
-------

- The authors need to use better and consistent notations. Also, as the overall approach uses similar flow as previous papers such as DeepCoevolve, it is recommended that the authors make the presentation simpler to position it clearly with existing works. On page 3, section 3.2 both bold-face and normal letters are used as vectors. Is $\hat{x}_{u(t)}$ a vector?

- Please provide numbers to equations for better referencing

[1] Predicting Dynamic Embedding Trajectory in Temporal Interaction Networks, Kumar et. al. KDD 2019

**Experience Assessment:**

I have published one or two papers in this area.

**Review Assessment: Checking Correctness Of Derivations And Theory:**

N/A

**Review Assessment: Checking Correctness Of Experiments:**

I carefully checked the experiments.

**Review Assessment: Thoroughness In Paper Reading:**

I read the paper thoroughly.

---

> ### Author Response · Authors · 2019-11-13
> **Response to Review 3**
>
> Thank you for your detailed comments and suggestions.
> we first answer your comments about DIP model itself.
> Q1:     Do you also update cell states with selection mechanism? The DIP-UNIT equation in selection section does not show that update. Also, are the embeddings updated only during train or also during validation/evaluation?
>
> ANS1: we enhance the j-th layer dynamic representation and cell states by weighting the (j-1)-th hidden states(i.e. input for j-th layer) but not the cell states in (j-1)-th layer in current version ( where j=0 means input features). Please see our updated picture Fig.3 and Equation 7.  The embedding is updated for every event during train/validation/evaluation(using the model learned on training data).
>
> Q2:     The use of proposed Algorithm 2 is not well justified. (Part A:)Why does the author need coloring and hashing mechanism instead of simpler BFS/random walk routine to collect previous interactions?(Part B:) Also, is this subgraph created for each event or it is computed offline during training? (Part C:)Further, the subgraph used for selection mechanism same as subgraph used for backtracking in LSTM?
>
> ANS2: It is a good question.  Please see our updated version with a more clear toy example in Fig.2. The pipeline of our work is as follows :
> 1. Propose a temporal dependency graph(TDG) concept to depict complex dependencies among time-evolving interactions.
> 2. Learn representation using DIP units, i.e. ,graph-structured LSTM with time gates to handle irregular time intervals.
> 3. Considering computational burden, when an interaction(event) happened, we only use its k-depth subgraph to compute their new representations(Fig2.d) which is similar to chain-LSTM training unfolded with max k steps and k-hops setting in static graph.
> 4. When updating dynamics of an event,  how to obtain k-depth subgraph is a practical problems.  The purpose of coloring method is to find TDG dependencies among interactions and  then we can construct k-depth TDG incrementally for each event.  So the coloring operation is a pre-step for constructing TDG graph.
> As for your question,  A. The bfs/random walk work only on a graph which already exists, but in our situation we don't  know what TDG is (as graph is growing up) and Coloring is a pre-step to help construct TDG.  B. Yes, each event needs its temporal subgraph information on TDG and we can train it in parallel. This is quite different from dynamic methods like deep coevol, JODIE which use RNN-like equations to update nodes states incrementally and thus have parallel and efficiency problems when processing training data with a long time duration. C. The selection methods utilize "mutual information" between two subgraphs of interactive nodes to adjust importance of nodes. All the calculation is based on k-depth temporal dependency graph.  The subgraph used for selection mechanism is of course the same as subgraph used for backtracking in LSTM.
>
> Q3:  is it true that the training is done in order of ColorGraphSeq or is it done in order of dataset? How does the authors capture dependencies across dataset in later case?
>
> ANS3:  As answered in ANS2, once we obtain the k-depth subgraph for an event(which means we only consider max k-depth history information as we do in chain-lstm),  we can train it parallelly. The purpose of ColorGraphSeq is only a pre-step for constructing a TDG  from a collected training data.   For new datasets, we can first incrementally update and find  dependencies between new data and old data(similar to  line 3 to 14 in Algorithm 1) and then get updated k-depth subgraphs for new interactions.
>
> Q4: How does the scaling parameter and alpha affect the performance and what are their roles?
>
> ANS4:  The purpose of fusion is to utilize all the dynamic representation of nodes at different Layers(at different time scales) to give a final representation.  The alpha is a learnt parameter used to weight node representation at different layers and then sum them.  Also the scaling parameter is a learnt parameter and aid the optimization process similar to Elmo

---

> > ### Author Response · Authors · 2019-11-14
> > **Response to Review 3-II**
> >
> > we now answer your comments about DIP experiments.
> > In our updated version, we provide a comparison with your suggested baseline  JODIE and provide results analysis in 5.3.2 and 5.4.2. Meanwhile we investigate the effects of different k and L on interaction prediction and interaction classification(Appendix B.3).
> >
> > Q5:  "The authors include support for new nodes for interaction classification task but remove them for interaction prediction task which is strange. Is there a specific reason for this? What is the effect on the performance if new nodes are allowed in test? Further, why is interaction classification not compared with temporal baselines? All baselines produce embeddings and the authors mention that classification for this paper is independent of marker history. While the temporal baselines do not train for the task, the authors can train a second stage classifier with learned embeddings to perform classification"
> >
> > A5: yes, there is a specific reason that we don't compare all the baselines in the interaction classification task: a. the dataset in this task  has a lot of unseen nodes so the transductive method like CTDNE can't fit  in this task. b. all the baseline dynamic methods are only unsupervised version while our method,GCN and gbdt are end-to-end supervised methods
> >
> > Q6: it seems datasets in experiments does have non-bipartite case? Is this true or the method only works for bipartite case?
> >
> > A6: Our method is built for modeling dynamic interactions. Although the datasets using in our experiments are heterogeneous graphs, it could be naturally applied to isomorphic graph such as citation graph. (Dynamic citation behaviors will construct the temporal dependency graphs.)

---

### Decision · Program_Chairs · 2019-12-19

**Decision:**

Reject

**Comment:**

All reviewers rated this paper as a weak reject.
The author response was just not enough to sway any of the reviewers to revise their assessment.
The AC recommends rejection.